# Horned Melon Pulp, Peel, and Seed: New Insight into Phytochemical and Biological Properties

**DOI:** 10.3390/antiox11050825

**Published:** 2022-04-23

**Authors:** Olja Šovljanski, Vanja Šeregelj, Lato Pezo, Vesna Tumbas Šaponjac, Jelena Vulić, Teodora Cvanić, Siniša Markov, Gordana Ćetković, Jasna Čanadanović-Brunet

**Affiliations:** 1Faculty of Technology Novi Sad, University of Novi Sad, Bulevar cara Lazara 1, 21000 Novi Sad, Serbia; oljasovljanski@uns.ac.rs (O.Š.); vesnat@uns.ac.rs (V.T.Š.); jvulic@uns.ac.rs (J.V.); sinisam@uns.ac.rs (S.M.); gcetkovic@uns.ac.rs (G.Ć.); jasnab@uns.ac.rs (J.Č.-B.); 2Institute for General and Physical Chemistry, Studenski trg 12/V, 11000 Belgrade, Serbia; latopezo@yahoo.co.uk

**Keywords:** horned melon, fruit waste, multiobjective optimization, antioxidative activity, antimicrobial activity, artificial neural network

## Abstract

Artificial neural intelligence was established for the estimation, prediction, and optimization of many agricultural and food processes to enable enhanced and balanced utilization of fresh and processed fruits. The predictive capabilities of artificial neural networks (ANNs) are evaluated to estimate the phytochemical composition and the antioxidant and antimicrobial activity of horned melon (*Cucumis metuliferus*) pulp, peel, and seed. Using multiobjective optimization, the main goals were successively achieved through analysis of antimicrobial potential against sensitive microorganisms for peel (*Bacillus cereus*, *Pseudomonas aeruginosa*, *Aspergillus brasiliensis*, and *Penicillium aurantiogriseum*), pulp (*Salmonella enterica* *subsp. enterica* serotype Typhimurium), and seed samples (*Saccharomyces cerevisiae* and *Candida albicans*), and its connection with phytochemical and nutritional composition and antioxidant activity. The highly potent extracts were obtained from peels which represent a waste part with strong antioxidant and antifungal capacity. Briefly, the calculated inhibition zone minimums for sensitive microorganisms were 25.3–30.7 mm, while the optimal results achieved with carotenoids, phenolics, vitamin C, proteins, lipids, DPPH, ABTS, and RP were: 332.01 mg β-car/100 g, 1923.52 mg GAE/100 g, 928.15 mg/100 g, 5.73 g/100 g, 2.3 g/100 g, 226.56 μmol TE/100 g, 8042.55 μmol TE/100 g, and 7526.36 μmol TE/100 g, respectively. These results imply the possibility of using horned melon peel extract as an antioxidant and antifungal agent for food safety and quality.

## 1. Introduction

Nowadays, food demands are increasing rapidly; additionally, awareness of the importance of a healthy diet is spreading worldwide. Therefore, the consumption and production of fruits and vegetables are growing [1], and those trends in the food industry are resulting in large amounts of waste that account for one-third of produced food [2]. Waste accumulation is a serious ecological and economical problem, and a potential solution is the reuse of plant-based waste as a rich source of phytochemicals, with antimicrobial and antioxidant effects [3]. In recent decades, the phytochemicals naturally present in waste and non-edible parts of plants, fruits, and vegetables have been reintroduced into the food chain as natural food additives [4]. Additionally, the utilization of novel ingredients is very popular both among the scientific community and food and pharmaceutical industries. Fruits contain chemical compounds responsible for specific flavors and qualities which determine their appearance, nutritional value, and food security [5]. Even though it is characteristic for plants to produce a large number of secondary metabolites, diversity is detected within it and there is evidence that this phenomenon happens to allow the plant to adaptively respond to the different impacts [6].

Horned melon (*Cucumis metuliferus* E. Mey. Ex. Naudin) is a fruit from the Cucurbitaceae family. It is originally from Africa, but now it has expanded worldwide as a luxury fruit due to its interesting appearance and the possibility of long storage [7]. The mature fruit has green flesh, rich with white seeds, inside of orange-yellow bark covered with strong spiny outgrowths [8]. Even though all parts of the fruit are edible, the majority of people eat only the inner parts and the unwanted parts are thrown away, but this waste has potential for implementation in food technology. Due to the presence of bioactive molecules, such as flavonoids, glycosides, phenolic compounds, and carotenoids, it can have antifungal, antimicrobial, antidiabetic, antihypertensive, and antioxidant effects [2].

In the past decade, the opportunity to study and use artificial neural networks (ANN) in a variety of applications makes it an essential tool in the development of food products and the utilization of food waste. This mathematical tool, simulating human brain interactions, enables advanced modelling and prediction of targeted characteristics in engineering and technological processes [9,10,11]. Briefly, ANNs can be used in food safety and quality analysis, such as microbial growth, presence of contaminations, and/or antimicrobial agents, and can also be used for predicting physical, chemical, functional, and sensory properties of various agricultural products during food processing, distribution, and storage [11]. Although ANN modelling is widely applied in fruit safety and quality processing, to the authors’ knowledge, there is no extant use of these mathematical models in horned melon analysis.

Taking all the aforementioned information into consideration, this research aimed to investigate the nutritive, antioxidant, and antimicrobial potentials of *Cucumis metuliferus*, originating from the Pannonia plain in the Republic of Serbia, for the first time. The innovation of this study is reflected not only in the potential implications for plant origin from a specific, non-native area, but also in the artificial neural network concept of multiobjective optimization of the most potent parts of fruit waste, with the potential uses as antioxidant and antimicrobial agents.

## 2. Materials and Methods

### 2.1. Fruit Samples

Horned melon fruits were produced as organic fruits by agricultural holding at the Fruška gora Mountain, in the area of Pannonia plain, Novi Sad, Republic of Serbia (45°12′ N, 19°45′ E). The five mature fruits were randomly selected for analysis (Figure 1). The seed, pulp, and peel were separated manually in aseptic laboratory conditions and stored after being freeze-dried. Briefly, all samples were frozen at −40 °C for 2 h in a Martin Crist Alpha 2–4 (Osterode, Germany) freeze-drier. The main drying process was performed at a pressure of 0.01 mbar and temperatures from −40 to 20 °C for 48 h. The final drying lasted 4 h at pressure of 0.005 mbar and temperatures from 20 to 30 °C. The collected freeze-dried samples were stored at −20 °C until further use. Freeze-dried seed, pulp, and peel of each sample were manually ground (Appendix A) using a laboratory mortar and pestle (Lab Logistics Group GmbH, Meckenheim, Germany). The grounding process was conducted until coarse powder was obtained. Powdered samples were packed in plastic boxes and stored at −20 °C in freezer LGUEX 1500 (Lab Logistics Group GmbH, Meckenheim, Germany) till extraction.

### 2.2. Preparation of Horned Melon Extracts

For analysis of total phenolic and carotenoid contents, as well as antioxidant activity, freeze-dried samples of horned melon pulp, peel, and seed were extracted according to the procedure of Šeregelj et al. [12]. Briefly, freeze-dried samples were extracted three times using acetone/ethanol mixture (36:64) in a solid to solvent ratio of 1:20 for 10 min, with the same volume of solvents. The extraction was performed using a laboratory shaker at 300 rpm, under light protection, at room temperature. The three obtained extracts were filtered (Whatman paper No. 1), combined, and stored in dark bottles at −20 °C. For chlorophylls content, the method described by Aborus et al. [13] was applied. For analysis of vitamin C, the horned melon samples were extracted with 80% methanol aqueous solution containing 0.05% acetic acid in an ultrasonic bath (RK 52 H, 1.8 L, SA GR, Lab Logistics Group GmbH, Meckenheim, Germany) for 30 min at room temperature.

### 2.3. Phytochemical Analysis

Total phenolic content (TPh) in samples was established using the Folin–Ciocalteau spectrophotometric method adapted to microscale [14]. The obtained results were expressed as gallic acid equivalents (GAE) per 100 g of dried sample. The content of total carotenoids (TCar) was determined spectrophotometrically according to the method in [15], and the results were expressed as mg of β-carotene equivalent per 100 g of dried sample. The measurements of the content of total chlorophylls, chlorophylls a, and chlorophylls b were based on the spectrophotometric method detailed in Lichtenthaleru [16] and expressed mg chlorophyll equivalent per 100 g of dried sample. The vitamin C was analyzed by the HPLC method detailed in Vulić et al. [17]. HPLC results were expressed as mg of ascorbic acid per 100 g of dried sample. For all spectrophotometric analysis in this study was used UV-1800 UV/VIS spectrometer (Shimadzu, Kyoto, Japan).

### 2.4. Nutritive Analysis

The procedure for determining the fat content was performed by the Soxhlet extraction method [18] using Soxhlet Apparatus (DWK, Mainz, Germany) using diethyl ether as a solvent. The protein content of the samples was determined by the Kieladl method described in the study by Marcó et al. [19] and using Kjeldahl Nitrogen Analyzer (BKN-983, BIOBASE, Karnataka, India). All analyses were repeated three times.

### 2.5. Antioxidant Assays

The antioxidant capacity was assessed following different methods: 2,2-diphenyl-1-picrylhydrazyl (DPPH), and 2,2′-azino-bis-3-ethylbenzothiazoline-6-sulphonic acid (ABTS), as described by Tumbas Šaponjac et al. [20]; reducing power (RP) according to Oyaizu [21]. The antioxidant capacity was expressed as millimoles of Trolox equivalent (TE) per 100 g of dried sample.

### 2.6. Antimicrobial Assays

To observe the antimicrobial activity of the extracts, 11 microorganisms were involved in the disk diffusion testing method. Briefly, Gram-negative bacteria *Escherichia coli* ATCC 25922, *Pseudomonas aeruginosa* ATCC 27853, *Salmonella enterica subsp. enterica* serotype Typhimurium ATCC 13311 (in the following text as *S.* ser. Typhimurium), and Gram-positive bacteria *Bacillus cereus* ATCC 11778, *Staphylococcus aureus* ATCC 25923, *Enterococcus faecalis* ATCC 19433, and *Listeria monocytogenes* ATCC 35152 were used. As representatives of yeasts and fungi, *Saccharomyces cerevisiae* ATCC 9763 and *Candida albicans* ATCC 10231, as well as *Aspergillus brasiliensis* ATCC 16404 and *Penicillium aurantiogriseum* ATCC 16025, were selected. As Mićić et al. [22] reported, Mueller–Hinton or Sobouraud maltose agar was inoculated with microbial suspensions (approx. 6 log CFU/mL), while the extract (15 μL, 50 mg/mL) was applied onto three sterile discs (ø 6 mm). As a negative control, sterile distilled water was used, while positive controls were clavulanic acid (Sigma-Aldrich, St. Louis, MI, USA) and cycloheximide (Acros Organic, NJ, USA). The obtained results were interpreted as the inhibition zones, as follows: sensitive (diameter of the zone above 26 mm), intermediary (diameter 22–26 mm), and resistance (diameter below 22 mm). The minimal inhibitory concentration (MIC) was evaluated for all microorganisms that are sensitive to the selected extracts using different concentrations of the extract (50–0.39 mg/mL). The number of microorganisms in treated samples (Nt) was compared with the number of microorganisms in the blank sample (Nc) and the MIC was calculated using Equation (1).
MIC = (Nc − Nt)/Nc · 100 (%)(1)

### 2.7. Artificial Neural Network (ANN) Modelling

The artificial neural network (ANN) model for predictive purposes was developed in the form of a multilayer perceptron (MLP). Prior to ANN calculation, input and output data were normalized by applying min–max normalization [23]. The input data are loaded to the ANN inputs in the ANN calculation [24,25]. The training process was repeated 100,000 times, investigating different ANN topologies, with a different number of neurons in hidden and output layers (5–20), different activation functions, and initial different weight coefficients and biases values. The Broyden–Fletcher–Goldfarb–Shanno (BFGS) algorithm was utilized for finding solutions to the unconstrained nonlinear optimization [23]. Coefficients associated with the hidden and output layer of the ANN model (weights and biases) are grouped in matrices *W*_1_ and *B*_1_, and *W*_2_ and *B*_2_, respectively [26]:(2)$$Y=f_{1}(W_{2}\cdot f_{2}(W_{1}\cdot X+B_{1})+B_{2})$$

The biases and weight coefficients related to the hidden and the output layers of the model are represented in the matrices and vectors *W*_1_ and *B*_1_ and *W*_2_ and *B*_2_, respectively [27]. *Y* represents the output value, *f*_1_ and *f*_2_ represent the transfer function in the hidden and output layer, *X* represents the matrix of the input layer [28].

### 2.8. Global Sensitivity Analysis

Yoon’s global sensitivity equation was used to calculate the relative impact of the input parameters, according to the weight coefficients of the ANN models [29]:(3)$$RI_{ij}(\%)=\frac{\sum_{k=0}^{n}\left(w_{ik}\cdot w_{kj}\right)}{\sum_{i=0}^{m}\left|\sum_{k=0}^{n}\left(w_{ik}\cdot w_{kj}\right)\right|}\cdot 100\%$$
where *RI*—relative impact; *w*—weight coefficient in ANN model; *i*—input variable; *j*—output variable; *k*—hidden neuron; *n*—number of hidden neurons; *m*—number of inputs.

### 2.9. Multiobjective Optimization

The developed ANN model was used for the multiobjective optimization (MOO) with an intention to extract the most potent extract(s) which will result in a maximal antioxidant and antimicrobial potential. The solution of MOO was presented as a Pareto front, using the genetic algorithm (GA), applying the mutation, selection, inheritance, and crossover. The initial population was randomly generated, while the next generations were calculated by distance measure and non-dominated ranking of individual points [30].

### 2.10. Statistical Analyses

The data were processed statistically using the software package STATISTICA 10.0 (StatSoft Inc., Tulsa, OK, USA). All determinations were made in triplicate, all data were averaged, and expressed by mean values. The principal component analysis (PCA) was used to discover correlations among measured parameters, and to classify objects.

## 3. Results and Discussion

Phytochemical composition (total phenolics, carotenoids, and chlorophylls) are evaluated using several methods recommended by Aćimović et al. [31], while protein and lipid content was determined using well-known methodology. Additionally, the antimicrobial activity of the extracts was tested in vitro on selected bacterial and fungal strains from the American Type Culture Collection (ATCC). In accordance with the experiment setups, a test of positive and negative controls was performed (Appendix A), and the gained results were in line with expectations—the tested bacteria were sensitive in the presence of clavulanic acid, while actidione, as an antimycotic agent, inhibited the growth of yeast and fungal strains.

### 3.1. C. metuliferus Pulp Extracts

The obtained results for horned melon pulp extracts are presented in Table 1. The gained values for total phenolics, carotenoids, and chlorophylls are found to vary between 58.22–74.90 mg GAE/100 g, 0.81–1.14 mg β-car/100 g, and 4.23–5.06 mg/100 g, respectively. Additionally, all tested samples have a lower amount of chlorophyll a (~1.41 mg/100 g) than chlorophyll b (~2.78 mg/100 g). It is noticeable that pulp extract sample 5 has the highest concentration of carotenoids and phenolics. Matsusaka and Kawabata [32] have reported a lower yield of phenolics in pulp (2.0 mg GAE/g dry weight), while the total carotenoids content had 0.88 μg β-car/g dry weight, which is in correlation with the obtained results in this study. According to Vieira [7], chlorophylls in horned melon pulp are present in a minimal concentration, but the obtained range for chlorophylls in the tested samples (Table 1) indicated the presence of certain amounts of these pigments. Vitamin C is not detected in pulp samples, while Vieira et al. [7] reported that pulp can contain the mentioned vitamin in the concentration of approx. 5.3 mg/g of dry weight.

As shown in Table 1, the total amount of proteins in pulp varies from 2.05 to 3.45 g/100 g, while lipid content is between 0.70 and 0.97 g/100 g. The obtained values for protein concentration are higher than the values reported by Ferrara [8], which indicated 1.80 g/100 g of proteins in pulp samples. On the other hand, Sodipo et al. [33] reported a higher amount of lipids (1.26 g/100 g) compared with the obtained values in this study. The pulp samples characterized a lower DPPH activity, which varies between 60.13 and 85.81 μmol TE/100 g, compared with the results of 14.4 μmol TE/g stated by Matsusaka and Kawabata [32]. Conversely, the obtained ABTS activity of approx. 2421.54 μmol TE/100 g is higher than 13.0 μmol TE/g, reported by the previously mentioned authors. The study of Arrieta et al. [34] reveals lower results of reducing power compared with the results presented in Table 1, where aquatic and ethanolic extracts had reducing power values between 161.3 and 241.1 mmol TE/100 g.

Furthermore, Table 1 is also emphasized antimicrobial testing results for pulp extracts. It is noticed that the selected fungi and yeasts, as well as Gram-positive and Gram-negative bacteria, are not sensitive to the effect of horned melon pulp extracts. The appearance of an inhibition zone with a diameter of about 18.5 mm occurred only in the case of incubation of *S.* ser. Typhimurium. The obtained results partially agree with the results of Usman et al. [35], who detected zones of inhibition about 14 mm in diameter when examining the growth of *S*. ser. Gallinarum at different concentrations (200–1000 mg/mL) of methanolic extracts of pulp.

Additionally, the PCA analysis of the presented data for pulp samples explained that the first two principal components explained 71.83% of the total variance (43.17% and 28.66%, respectively) in the eleven variables space (Figure 2). Considering the results of the PCA analysis, *S*. ser. Typhimurium (which contributed 15.1% of the total variance), while the content of chlorophyll a (18.3%), total chlorophyll (15.3%), chlorophyll b (13.7%), reducing power (10.4%), and lipids content (8.8%) exhibited a negative influence on PC1. The positive contribution to PC2 calculation was observed for phenolics (14.5% of the total variance), DPPH (12.8%), and carotenoids (26.6%), while negative scores on PC2 calculation were observed for reducing power (10.2%), lipids content (14.9%), and ABTS (15.1%).

### 3.2. C. metuliferus Peel Extracts

The gained results of total phenolics (Table 2) show that peel extracts samples contain between 1728.94 and 1923.52 mg GAE/100 g, which implicates higher results than the 5.0 mg GAE/g dry weight reported by Matsusaka and Kawabata [32]. The presents of chlorophylls are not detected in any sample, while carotenoids reach the highest concentration in sample 2 (341.01 mg β-car/100 g). Vitamin C levels vary in peels from 467.53 to 928.15 mg/100 g. Analogous results are represented within a concentration of vitamin C around 3.44 mg/g [36]. The values of DPPH activity are between 158.13 and 226.56 μmol TE/100 g, while a higher value (14.2 μmol TE/g) is reported by Vieira [7]. Evaluation of ABTS activity in all samples demonstrates stronger activity than the previously reported results of 12.5 μmol TE/g [7]. The reducing power of peel is approx. 5943.95 μmol TE/100 g, while Vieira [7] reported a reducing activity of peel of 50%. In this study, concentrations of proteins and lipids in peel samples are approx. 5.73 g/100 g and 2.13 g/100 g, respectively, while Achikanu et al. [36] reported 2.95% proteins and 8.89% lipids, respectively.

Based on the results shown in Table 2, the antimicrobial activity of the peel samples was not manifested against two Gram-positive bacteria—*S. aureus* and *E. faecalis*—and two Gram-negative bacteria *E. coli* and *S*. ser. Typhimurium. Peel samples have inhibited the growth of one Gram-positive bacterium, *B. cereus*, and one Gram-negative bacterium, *P. aeruginosa*. Briefly, peel sample 5 had the weakest effect, with an inhibition zone diameter of 21 ± 1 mm for *B. cereus* strain and 27.3 ± 3 mm for *P. aeruginosa* strain. In contrast, sample 3 had the strongest effect, and the detected zone of 31.7 ± 0.6 mm for *B. cereus*, while the zone for *P. aeruginosa* of 37 ± 3 mm was measured. In summary, analyses of methanolic extracts of horned melon peel also show antimicrobial activity, but not against all used bacterial species. The choice of solvent also changed the strength of the antimicrobial action.

Therefore, the absence of the inhibition zone in *S. aureus*, *E. faecalis*, *E. coli*, and *S*. ser. Typhimurium may be due to inadequate solvent selection, and chemical compounds could not be isolated in sufficient quantities to inhibit these bacteria. In analyses performed by Aliero and Gumi [37], horned melon extracts did not show significant antimicrobial activity against the following bacteria *E. coli*, *B. subtilis*, and *P. aeruginosa*, while aqueous extract caused growth inhibition of *S. aureus*. According to the absence of yeast growth with the maximum obtained zone in the study (larger than 40 mm), it can be concluded that peel extracts have a very strong antimycotic effect. A similar effect was observed for tested fungi. *P. aurantiogriseum* was more sensitive than *A. brasiliensis*, while the widest observed inhibitory zones of inhibition were 31.1 ± 0.6 and 25.7 ± 0.6, respectively. Nwadiaro et al. [38] tested aqueous and ethanolic peel extracts against the following strains: *Aspergillus flavus*, *Fusarium oxysporum*, *Mucor* sp., *Penicillium citrinum*, and *Rhizopus stolonifer*. The results showed that the strains of the genera *Aspergillus* and *Penicillium* were the most sensitive. It is also noticeable that peel sample 3 has the strongest effect against all susceptible microorganisms. Based on Table 2, the MIC value for Gram-positive bacterium *B. cereus* and fungi strains *A. brasiliensis* and *P. aurantiogriseum* is 25 mg/mL, while for tested yeast *S. cerevisiae* and *C. albicans*, as well as for Gram-negative *P. aeruginosa*, the value is 3.125 mg/mL. In a study by Usman et al. [35], the MIC value for methanol extract was 50 mg/mL in the case of *S*. ser. Gallinarum strain; meanwhile, within this study, it was obtained that for *S*. ser. Typhimurium strain MIC value over > 50 mg/mL, which suggest differences in serotype level. According to research by Rakholiya et al. [39], most peek extracts of various fruits and vegetables had a pronounced effect on bacteria of the genus *Bacillus*, such as *B. subtilis* and *B. megaterium*, and on *B. cereus*, which correlates with the results of this study. Additionally, the same research showed that the peel has a stronger antibacterial than antifungal effect, while Aliero and Gumi [37] reported a greater antifungal effect. Namely, the results indicated that peel extracts have an antifungal effect against *Aspergillus flavus*, *Fusarium solani*, *Trichophyton mentagrophyte*, and *Microsporum canis*, but not on *A. niger*. In a study by Saleem and Saeed [40] who analyzed the antimicrobial properties of secondary raw materials of orange, lemon, horned melon, and banana, only horned melon extracts showed a pronounced effect on yeasts and fungi.

The PCA of the presented data for peel samples explained that the first two principal components explained 72.33% of the total variance (46.75% and 25.58%, respectively) in the twelve variables space (Figure 3). Considering the results of the PCA analysis, proteins content (which contributed 12.0% of the total variance), vitamin C (14.4%), DPPH (16.2%), ABTS (15.9%), and phenolics (15.1%) exhibited negative scores according to the first principal component (PC1). The positive contribution to PC2 calculation was observed for *A. brasiliensis* (8.8% of the total variance), while negative scores on PC2 calculation were observed for reducing power (24.0%), *P. aurantiogriseum* (22.3%), *B. cereus* (17.8%), and *P. aeruginosa* (7.9%).

### 3.3. C. metuliferus Seed Extracts

Table 3 summarizes the obtained results for the seed extract samples. The total amounts of carotenoids and phenolics are approx. 0.53 mg β-car/100 g and 141.25 mg GAE/100 g, respectively. Contrarily, Sadou et al. [33] claimed that the seed sample has 130 mg/g dry weight of β-carotene, which is a greater value compared with the obtained values in this study. On the other hand, total phenolics have been reported to have a higher concentration reaching 4.0 mg GAE/g in seed samples [32]. No detectable number of chlorophylls was found in the peel, nor in the seed samples, nor was it recognizable in the literature. Similar to pulp samples, in seed samples, vitamin C is not detected, but it is documented that the seed can contain about 2 mg/g of vitamin C [36]. Even though Bolek [1] has obtained high DPPH and ABTS activity of seed, 20.88 and 185.36 μmol TE/g, respectively, the results that are given in Table 3 show significantly lower activity. Additionally, reducing power values vary from 305.73 to 373.54 μmol TE/100 g, while Matsusaka and Kawabata [32] obtained a reducing power of around 8%. Sadou et al. [33] have reported that horned melon seed contains 23.8 g/100 g lipids and 23.2 g/100 g proteins which are in agreement with the results present in Table 3.

It is noticeable that seeds do not have antibacterial activity. Additionally, resistance was detected in both tested fungi strains. The antimicrobial effect of seeds, except for sample 5, was manifested against yeasts *S. cerevisiae* and *C. albicans*. Seed sample 1 had the strongest effect according to *S. cerevisiae*, where the inhibition zone with a diameter of 33 ± 4.4 mm was measured, while the same sample had a lower antimicrobial effect against *C. albicans*, where an enlightenment zone with a diameter of 23.33 ± 5 mm was observed. Furthermore, the same values of minimum inhibitory concentration (MIC) were recorded for all tested seed samples and were 6.25 mg/mL. Based on the obtained results of antimicrobial activity (Table 3), it can be concluded that the mixture of the present phytochemicals in the seed in the given concentrations does not exhibit an inhibitory effect against bacteria and fungi.

On the other hand, the inhibitory effect on yeast can be reflected in the presence of a certain concentration of tannins, which have been repeatedly shown to exhibit antimicrobial activity against species of the genus *Candida*: *C. albicans* ATCC 10231, *C. krusei* ATCC 34135, *C. glabrata* ATCC 2001, and *C. tropicalis* ATCC 28707 [41,42]. Based on the ATCC culture collection specification, *C. albicans* ATCC 10231 and *C. tropicalis* ATCC 28707 are strains which are highly resistant to a large number of chemical agents that are defined as antimicrobial agents. The mechanism of action of phenolics as an antifungal agent refers to the possibility of binding to ergosterol present in the structure of the cell membrane [42]. More precisely, the antifungal effect is manifested by binding to ergosterol, an essential lipid of the yeast cell membrane, leading to a violation of cell integrity. In addition to direct binding to ergosterol, the effect on ergosterol biosynthesis is significant because they reduce the amount of ergosterol present in the cell by reducing the expression of the gene encoding the protein involved in the biosynthesis of this lipid [43]. Based on the obtained results, it can be concluded that horned melon seed acts as a biocide agent against yeasts, and in order to better understand its potential use as a source of fatty acids and bioactive components, it is necessary to perform comprehensive chemical characterization.

The PCA of the presented data for seed samples explained that the first two principal components explained 83.97% of the total variance (55.99% and 27.08%, respectively) in the nine variables space (Figure 4). Considering the results of the PCA analysis, DPPH (which contributed 17.2% of the total variance), reducing power (16.9%), ABTS (16.2%), and phenolic (13.2%) contributed positively to the PC1 coordinate, while *C. albicans* (12.9%), *S. cerevisiae* (14.4%), and carotenoids content (7.5%) exhibited a negative influence on PC1. The negative contribution to PC2 calculation was observed for carotenoids (8.9% of the total variance), lipids (29.2%), proteins content (33.7%), phenolics (8.4%), and ABTS (7.1%).

From phytochemicals compound’s view, one of the crucial directions in understanding correlation of plant phytochemicals in human diet is the possibility of present bioactives to modify the gut microbiota and consequently host health [44]. Briefly, the presence of phytochemicals in food can be correlated with alteration and colonialization of gut microbiota, which deepens on dosage of phytochemicals, timing, and daily diet routine, as well as activity and concentration of microbes [45]. Therefore, modern researchers are investigating the bioactivity and bioavailability of functional compounds in the concentration which will be used in food matrix. Consequently, future investigation of selected horned melon extracts needs to be direct in this comprehensive study approach, involving the tested food-borne microorganisms in this study, but also representatives of gastrointestinal tract. Investigation of synergetic effects between phytochemicals, microbial cells, and release kinetics is imperative for understanding the bioavailability of targeted phytochemicals [46].

### 3.4. ANN Model

The ANN was developed to predict the phytochemical composition of peel, pulp, and seed samples (phenolics, carotenoids, chlorophylls (total, a, and b), and vitamin C), nutritional composition (proteins and lipids), antioxidant activity (DPPH, ABTS, and RP), and antimicrobial activity (the inhibition zone and MICs). According to ANN performance, the optimal number of neurons for prediction of antimicrobial activity of peel samples (against *B. cereus*, *P. aeruginosa*, *A. brasiliensis*, and *P. aurantiogriseum*), pulp samples (*S*. ser. Typhimurium), and seed samples (*S.*
*cerevisiae* and *C.*
*albicans*) in the hidden layer were 12 (network MLP 8-12-4), 10 (network MLP 10-6-1) and 7 (network MLP 7-7-2), respectively, to obtain high values of R2 (overall 0.999) and low values of the sum of squares (SOS). The input parameters for ANN models were as follows: carotenoids, phenolics, vitamin C, proteins, lipids, DPPH, ABTS, and RP for the peel samples prediction model; carotenoids, phenolics, chlorophylls, chlorophylls a, chlorophylls b, proteins, lipids, DPPH, ABTS, and RP for the pulp samples; and carotenoids, phenolics, proteins, lipids, DPPH, ABTS, and RP for the seed samples. The applied training algorithms were BFGS 39 (peel), BFGS 2458 (pulp), and BFGS 28, with exponential for hidden activation and identity for output activation function. ANN models were complex (160, 73, and 72 weight biases for peel, pulp, and seed samples) because of the high nonlinearity of the developed system [23].

### 3.5. Global Sensitivity Analysis—Yoon’s Interpretation Method

The influence of input variables carotenoids, phenolics, vitamin C, proteins, lipids, DPPH, ABTS, and RP on antimicrobial effect against *B. cereus*, *P. aeruginosa*, *A. brasiliensis*, and *P. aurantiogriseum* was studied, based on the Yoon’s interpretation method of a developed ANN model for peel samples. The graphical presentation of Yoon’s analysis for ANN model results is shown in Figure 5. The most positive influence on the antimicrobial effect of peel extract against *B. cereus* was observed for lipids content (relative influence was 20.17%), while the most negative influence was observed for proteins content (−25.4%), DPPH (−14.35%), and ABTS (−18.42%). According to Figure 5, the most positive influence on *P. aeruginosa* was observed for lipids content (relative influence was 17.24%) and RP (26.13%), while the most negative influence was observed for proteins content (−28.11%). The most positive influence on *A. brasiliensis* was observed for lipids content (relative influence was 22.53%) and RP (18.55%), while the most negative influence was observed for carotenoids (−16.27%) and proteins content (−18.76%). Additionally, the most positive influence on *P. aurantiogriseum* was observed for vitamin C content (relative influence was 18.43%) and lipids content (24.47%), while the most negative influence was observed for RP (−29.64%).

The influence of input variables: carotenoids, phenolics, chlorophylls, chlorophyll a, chlorophyll b, proteins, lipids, DPPH, ABTS, and RP on *S*. ser. Typhimurium was studied, based on the Yoon’s interpretation method of a developed ANN model for pulp samples. The graphical presentation of Yoon’s analysis for ANN model results is shown in Figure 6. The most positive influence on *S*. ser. Typhimurium was observed for DPPH (relative influence was 12.54%), ABTS (17.38%), and RP (8.74%), while the most negative influence was observed for chlorophyll a (−14.47%), chlorophyll b (−14.43%), proteins (−13.32%), and lipids content (−11.49%).

The influence of input variables: carotenoids, phenolics, proteins, lipids, DPPH, ABTS, and RP on *S. cerevisiae* and *C. albicans* was studied, based on Yoon’s interpretation method of a developed ANN model for seed samples. The graphical presentation of Yoon’s analysis for ANN model results was shown in Figure 7. The most positive influence on *S. cerevisiae* was observed for lipids content (relative influence was 44.62%), while the negative influence was observed for carotenoid content (−14.11%), DPPH (−18.13%), and ABTS (−13.50%). According to Figure 7, the most positive influence on *C. albicans* was observed for lipids content (relative influence was 33.28%), while the most negative influence was observed for DPPH (−21.55%).

### 3.6. Multiobjective Optimization of the Outputs of the ANN

The optimization of the ANN outputs was performed using results presented in Table 1, Table 2 and Table 3. One of the main goals of this investigation was to optimize the antimicrobial activity against sensitive microorganisms in peel samples (*B. cereus*, *P. aeruginosa*, *A. brasiliensis*, and *P. aurantiogriseum*), pulp samples (*S*. ser. Typhimurium), and seed samples (*S. cerevisiae* and *C. albicans*), according to the following input variables: carotenoids, phenolics, vitamin C, proteins, lipids, DPPH, ABTS, and RP for the peel samples prediction model; carotenoids, phenolics, chlorophylls, chlorophyll a, chlorophyll b, proteins, lipids, DPPH, ABTS, and RP for the pulp samples; and carotenoids, phenolics, proteins, lipids, DPPH, ABTS, and RP for the seed samples. These numerical tasks were solved separately for ANN models for peel, pulp, and seed samples, using the MOO calculation in Matlab. The MOO procedure was defined to find the minimums of output variables. Constrains used in the optimization procedure were applied within the experimental range of parameters. The number of generations reached 536 for the peel prediction model, 526 for pulp, and 262 for the seed model, while the size of the population was set to 100 for each input variable for all models. The number of points on the Pareto front was 23, 25, and 19 for peel, pulp, and seed samples models, respectively. The calculated minimums for *B. cereus*, *P. aeruginosa*, *A. brasiliensis*, and *P. aurantiogriseum* in peel samples were 30, 30.7, 25.3, and 29.3 mm, respectively. The optimal results were achieved with carotenoids, phenolics, vitamin C, proteins, lipids, DPPH, ABTS, and RP, with results of 332.01 mg β-car/100 g, 1923.52 mg GAE/100 g, 928.15 mg/100 g, 5.73 g/100 g, 2.3 g/100 g, 226.56 μmol TE/100 g, 8042.55 μmol TE/100 g, and 7526.36 μmol TE/100 g, respectively.

The calculated minimum for *S*. ser. Typhimurium in pulp samples was 18.3 mm. The optimal results were achieved with carotenoids, phenolics, chlorophylls, chlorophyll a, chlorophyll b, proteins, lipids, DPPH, ABTS, and RP, with results of 1.14 mg β-car/100 g, 74.9 mg GAE/100 g, 4.91 mg/100 g, 1.49 mg/100 g, 3.31 mg/100 g, 3.45 g/100 g, 0.83 g/100 g, 85.81 μmol TE/100 g, 2421.54 μmol TE/100 g, and 275.34 μmol TE/100 g, respectively. The calculated minimums for *S. cerevisiae* and *C. albicans* in peel samples were 26.0 and 20.0, respectively. Optimal results were achieved when carotenoids, phenolics, proteins, lipids, DPPH, ABTS, and RP were at measures of 0.62 mg β-car/100 g, 144.39 mg GAE/100 g, 25.12 g/100 g, 28.22 g/100 g, 58.4 μmol TE/100 g, 1753.29 μmol TE/100 g, and 368.34 μmol TE/100 g, respectively.

## 4. Conclusions

Although information about the nutritional characteristics of horned melon fruit exists, it is necessary to conduct more tests to investigate the nutritional, antimicrobial, and phytochemical characteristics of other parts of horned melon, such as peels or seeds, as highly potent waste parts. Therefore, this study used spectrophotometric methods for low-cost and rapid results of in vitro antioxidant potential, as well as in vitro antimicrobial testing methods for screening of the biological activities of five samples of peel, pulp, and seed extracts, respectively. On the other hand, some of spectrophotometric methods used have limited sensitivity and require additional testing. Therefore, further investigations into the use of highly sensitive high-performance liquid chromatography (HPLC) will ensure that precise and relevant information about different chemical compounds is gleaned. Additionally, advanced mathematical modelling can be a solution for better understanding of the connection between samples and the definition of the optimal samples for further investigation. According to the obtained results, peel extracts are the most promising source of phytochemicals that influence antioxidant and antimicrobial activity. Additionally, using advanced mathematical tools such as ANN modelling and multiobjective optimization, the second angle of research is to approach mutual relative influences on the tested biological activities of horned melon samples. These results imply the possibility of using horned melon peel extract as a potential antioxidant and antifungal agent for food safety and quality, which will be explored further in future investigations.

## Figures and Tables

**Figure 1 antioxidants-11-00825-f001:**
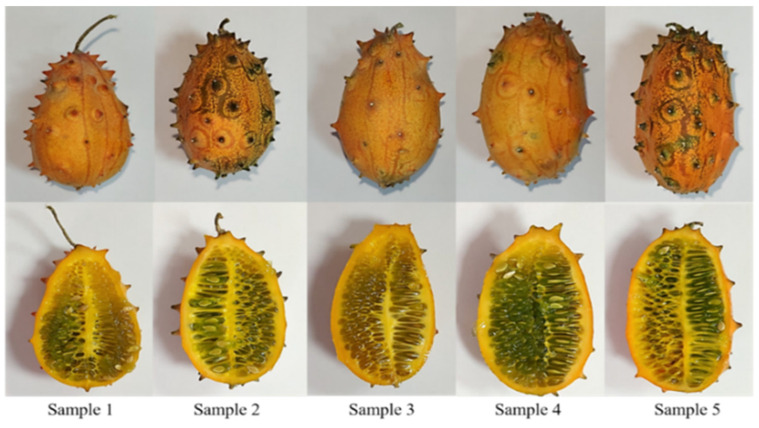
Horned melon samples.

**Figure 2 antioxidants-11-00825-f002:**
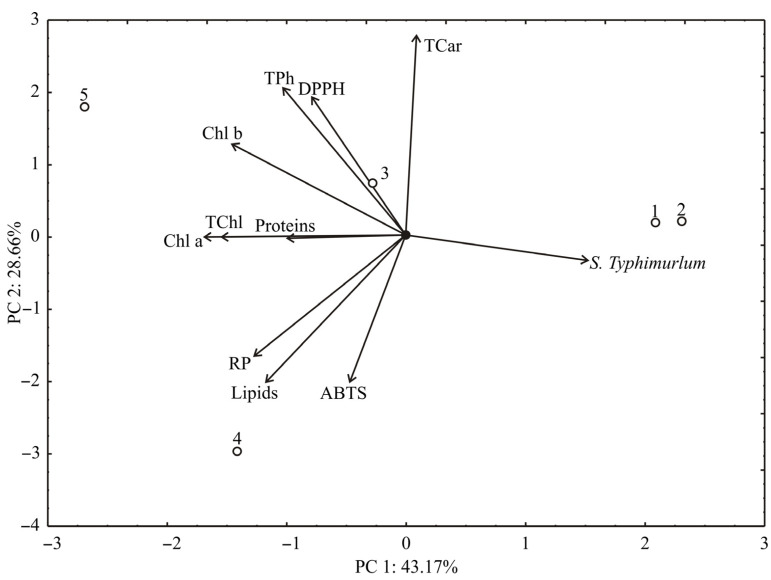
PCA ordination of variables based on correlations of pulp samples.

**Figure 3 antioxidants-11-00825-f003:**
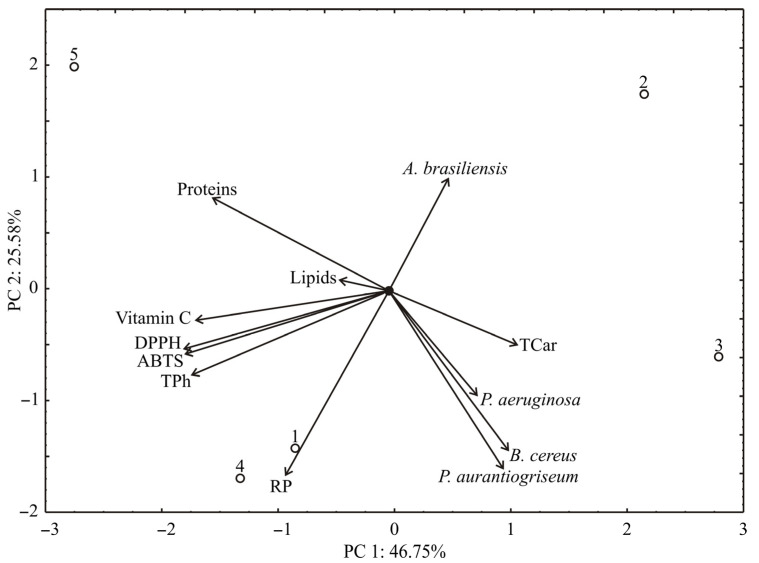
PCA ordination of variables based on correlations of peel samples.

**Figure 4 antioxidants-11-00825-f004:**
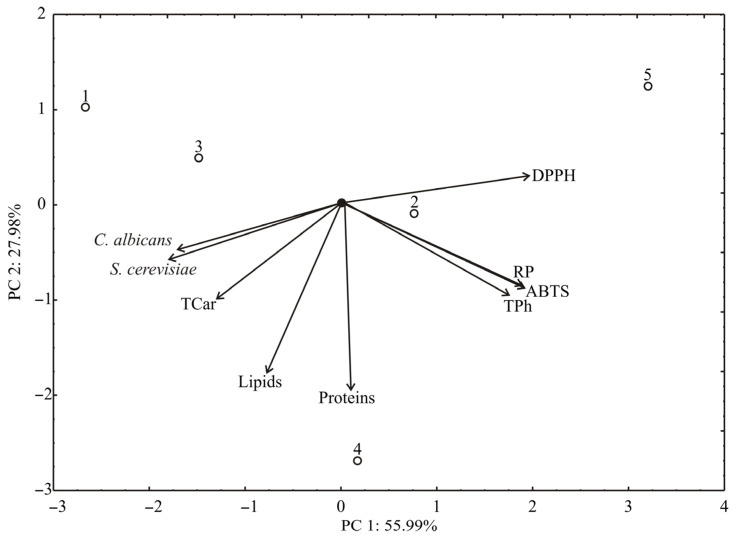
PCA ordination of variables based on correlations of seed samples.

**Figure 5 antioxidants-11-00825-f005:**
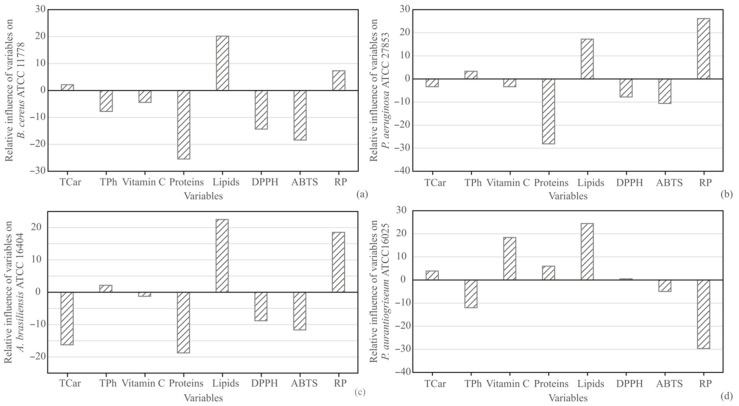
The relative importance of input variables (TCar, TPh, Vitamin C, Proteins, Lipids, DPPH, ABTS and RP) on (**a**) *B. cereus* ATCC 11778, (**b**) *P. aeruginosa* ATCC 27853, (**c**) *A. brasiliensis* ATCC 16404 and (**d**) *P. aurantiogriseum* ATCC 16025 in peel.

**Figure 6 antioxidants-11-00825-f006:**
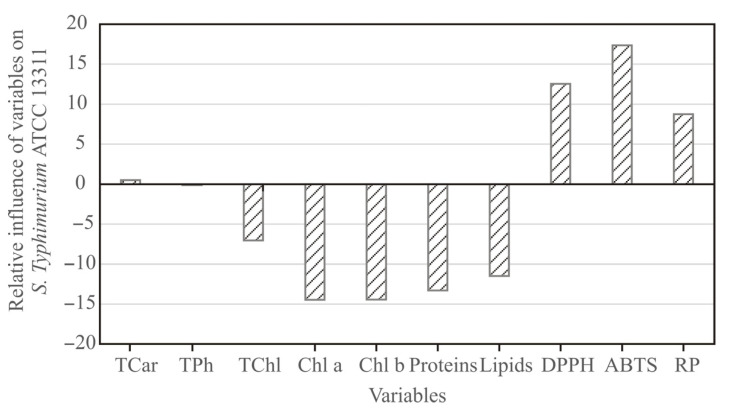
The relative importance of input variables on *S*. ser. Typhimurium in pulp.

**Figure 7 antioxidants-11-00825-f007:**
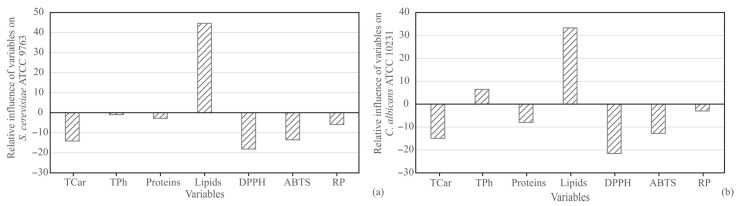
The relative importance of input variables on yeast strains in seed samples.

**Table 1 antioxidants-11-00825-t001:** The obtained results for pulp extracts.

Analysis	Unit	Sample				
		Pulp 1	Pulp 2	Pulp 3	Pulp 4	Pulp 5
Phytochemical Composition						
TCar	(mg β-car/100 g)	1.05 ± 0.04	1.02 ± 0.01	0.96 ± 0.02	0.81 ± 0.03	1.14 ± 0.01
TPh	(mg GAE/100 g)	64.51 ± 2.07	58.22 ± 7.97	61.90 ± 5.23	58.71 ± 5.11	74.90 ± 1.31
TChl	mg/100 g	4.23 ± 0.12	4.25 ± 0.04	5.06 ± 0.31	4.87 ± 0.65	4.91 ± 0.09
Chl a		0.91 ± 0.03	1.01 ± 0.29	1.43 ± 0.09	1.41 ± 0.09	1.49 ± 0.51
Chl b		2.41 ± 0.06	2.56 ± 0.31	3.25 ± 0.02	2.78± 0.18	3.31 ± 0.60
Vitamin C		nd				
Nutritional composition						
Proteins	g/100 g	2.61 ± 0.08	2.70 ± 0.13	2.05 ± 0.04	3.01 ± 0.06	3.45 ± 0.17
Lipids		0.71 ± 0.01	0.73 ± 0.01	0.70 ± 0.01	0.97 ± 0.02	0.83 ± 0.01
Antioxidant activity						
DPPH	μmol TE/100 g	77.98 ± 2.13	60.13 ± 7.01	70.15 ± 6.38	63.45 ± 8.22	85.81 ± 9.15
ABTS		2508.69 ± 23.14	2315.73 ± 14.01	2306.82 ± 9.22	2598.93 ± 14.12	2421.54 ± 25.26
RP		271.53 ± 11.22	260.99 ± 11.78	269.28 ± 1.99	284.03 ± 12.01	275.34 ± 9.86
Antimicrobial activity						
Inhibition zone (mm)						
*B. cereus*		nd				
*S. aureus*						
*E. faecalis*						
*E. coli*						
*P. aeruginosa*						
*S*. ser. Typhimurium		19.7 ± 2.1	19 ± 1	18.7 ± 0.6	18.7 ± 0.6	18.3 ± 1.1
*S. cerevisiae*		nd				
*C. albicans*						
*A. brasiliensis*						
*P. aurantiogriseum*						
Minimal inhibitory concentration (mg/mL)						
*B. cereus*		>50				
*S. aureus*						
*E. faecalis*						
*E. coli*						
*P. aeruginosa*						
*S*. ser. Typhimurium						
*S. cerevisiae*						
*C. albicans*						
*A. brasiliensis*						
*P. aurantiogriseum*						

nd—not detected.

**Table 2 antioxidants-11-00825-t002:** The obtained results for peel extracts.

Analysis	Unit	Sample				
		Peel 1	Peel 2	Peel 3	Peel 4	Peel 5
Phytochemical Composition						
TCar	(mg β-car/100 g)	330.88 ± 14.12	341.01 ± 4.28	326.68 ± 8.31	332.01 ± 13.01	312.51 ± 9.93
TPh	(mg GAE/100 g)	1897.20 ± 21.88	1728.94 ± 9.41	1758.81 ± 38.18	1923.52 ± 12.14	1899.23 ± 15.13
TChl	mg/100 g	nd				
Chl a						
Chl b						
Vitamin C		625.12 ± 11.17	467.53 ± 5.74	448.41 ± 9.26	928.15 ± 6.13	860.05 ± 5.20
Nutritional composition						
Proteins	g/100 g	6.16 ± 0.21	5.24 ± 0.29	5.27 ± 0.11	5.73 ± 0.10	8.19 ± 0.31
Lipids		1.74 ± 0.02	1.94 ± 0.07	2.13 ± 0.03	2.30 ± 0.06	2.20 ± 0.01
Antioxidant activity						
DPPH	μmol TE/100 g	211.53 ± 7.95	163.47 ± 30.57	158.13 ± 21.12	226.56 ± 9.59	219.17 ± 10.12
ABTS		7845.91 ± 57.13	5472.91 ± 67.19	5183.46 ± 22.28	8042.55 ± 31.06	7812.13 ± 54.89
RP		7337.34 ± 12.53	5393.22 ± 5.12	5943.95 ± 31.15	7526.36 ± 38.19	5892.16 ± 12.17
Antimicrobial activity						
Inhibition zone (mm)						
** *B. cereus* **		26 ± 0.0	25 ± 2.6	31.7 ± 0.6	30 ± 1.0	21 ± 1.0
** *S. aureus* **		nd				
** *E. faecalis* **						
** *E. coli* **						
*P. aeruginosa*		27.7 ± 2.5	25.3 ± 1.1	37 ± 3	30.7 ± 1.1	27.3 ± 3
*S*. ser. Typhimurium		nd				
** *S. cerevisiae* **		40.0 ± 0.0				
** *C. albicans* **						
** *A. brasiliensis* **		23 ± 1.0	25.7 ± 0.6	25.3 ± 2.1	25.3 ± 1.1	25.3 ± 1.1
** *P. aurantiogriseum* **		29.7 ± 0.63	27.3 ± 0.6	31.1 ± 0.6	29.3 ± 0.6	25.3 ± 1.1
Minimal inhibitory concentration (mg/mL)						
** *B. cereus* **		25				
** *S. aureus* **		>50				
** *E. faecalis* **						
** *E. coli* **						
** *P. aeruginosa* **		3.125				
***S*. ser.** Typhimurium		>50				
** *S. cerevisiae* **		3.125				
** *C. albicans* **						
** *A. brasiliensis* **		25				
** *P. aurantiogriseum* **						

nd—not detected.

**Table 3 antioxidants-11-00825-t003:** The obtained results for seed extracts.

Analysis	Unit	Sample				
		Seed 1	Seed 2	Seed 3	Seed 4	Seed 5
Phytochemical Composition						
TCar	(mg β-car/100 g)	0.59 ± 0.01	0.41 ± 0.03	0.53 ± 0.01	0.62 ± 0.01	0.45 ± 0.02
TPh	(mg GAE/100 g)	123.16 ± 5.12	141.25 ± 7.27	136.71 ± 9.36	144.39 ± 1.12	144.08 ± 5.74
TChl	mg/100 g	nd				
Chl a						
Chl b						
Vitamin C						
Nutritional composition						
Proteins	g/100 g	24.06 ± 1.21	23.97 ± 2.17	23.99 ± 1.07	25.12 ± 0.86	24.04 ± 1.88
Lipids		27.15 ± 0.06	27.68 ± 0.05	27.08 ± 0.11	28.22 ± 0.09	26.14 ± 0.02
Antioxidant activity						
DPPH	μmol TE/100 g	49.21 ± 1.15	74.52 ± 9.12	47.15 ± 3.31	58.40 ± 6.53	81.80 ± 10.13
ABTS		1425.96 ± 21.03	1706.52 ± 6.55	1516.86 ± 21.45	1753.29 ± 19.62	1777.09 ± 13.15
RP		305.73 ± 4.06	366.05 ± 6.76	330.85 ± 2.11	368.34 ± 1.45	373.54 ± 7.43
Antimicrobial activity						
Inhibition zone (mm)						
*B. cereus*		nd				
*S. aureus*						
*E. faecalis*						
*E. coli*						
*P. aeruginosa*						
*S*. ser. Typhimurium						
*S. cerevisiae*		33.0 ± 4.4	30.3 ± 2.1	29 ± 1.0	26 ± 1.0	nd
*C. albicans*		23.33 ± 5.0	25 ± 0.0	30 ± 4.0	20 ± 4.4	nd
*A. brasiliensis*		nd				
*P. aurantiogriseum*						
Minimal inhibitory concentration (mg/mL)						
*B. cereus*		>50				
*S. aureus*						
*E. faecalis*						
*E. coli*						
*P. aeruginosa*						
*S*. ser. Typhimurium						
*S. cerevisiae*		6.25				>50
*C. albicans*						
*A. brasiliensis*		>50				
*P. aurantiogriseum*						

nd—not detected.

## Data Availability

Data is contained within the article and the Appendix A.

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
