# Peer review of "Horned Melon Pulp, Peel, and Seed: New Insight into Phytochemical and Biological Properties"

_antioxidants, 2022, doi:10.3390/antiox11050825_

Round 1

Reviewer 1 Report

I think that the manuscript entitled “Horned melon pulp, peel, and seed: New insight into phytochemical and biological properties" deserves publication in the Antioxidants after major revision. The manuscript is interesting dealing with current research problems. However, it seems to me that I am little research subjects as for model research. Research methodologies should be described so that the experiments can be replicated.

Line 79: please complete information on the freeze-drying, incl. parameters

Line 80: please complete information “each sample were grounded into a coarse powder”, incl. equipment, parameters, city, country, size of powder

Lines 80-81: why is Figure S1 in the supplementary file and not in the body of the manuscript

Line 81: please complete information stored -20°C, incl. equipment, city, country

Lines 84, 96, 108, 115, 123: please provide information about the equipment used

Line 90 and another: please complete information about the “%”, “v/v or v/w or…”

Line 90: please complete information on the ultrasonic bath, incl. temp., name, city, country

Line 108: carbohydrates were analysis?

Line 127 and all the manuscript: “Typhimurium” “Gallinarum” why are they not italized, and begins a capital letter?

Line 190: Only “Results”?

Line 194: please change “in vitro” into “in vitro

Line 196: why is Table S1 in the supplementary file and not in the body of the manuscript

What extracts were they? Please describe the methodology in detail

Line 207: „(2.0 mg GAE/g dry weigh)” another units… in 100 g was 200 mg? it was higher

Line 208: “0.88 μg β-car/g fresh weight” no dry matter analysis

Line 213: „5.3 mg/g” dry or fresh weight?

Lines 233, 235, 240: microorganisms – italics

Line 241: please change “(18.3)” into “(18.3%)”

Figure 2, 3 and 4: why are more variables taken into account at the same time then the case? In my opinion, PCA is not the best statistical method for this system of experience and the amount of analyzed data.

Line 259: please change „(2021)” into “[…]”

Table 2, peel 3: only this sample developed antifungal activity against Sacharomyces cerevisiae and Candida albicans?

Line 286: please change „(2015)” into „[…]”

Line 337: please change „S. cerevisiae” into „S. cerevisiae

Line 337: “imbibition”?

Author Response

The authors would like to thank the Editor and Reviewer for a professional review as well as opportunity to make essential and crucial changes in our work. All Reviewer' remarks are accepted and paper is changed according to comments. The authors believe that the changed paper would satisfy the Reviewer' criteria and that it is going to be acceptable for publishing in the Antioxidants.

We decided to revise manuscript indicating with red letters all changes directly on the revised manuscript. Please find in the attachment the word file with a point-by-point responses to the comments.

Best regards,

Dr. Vanja Šeregelj and co-authors

Reviewer 2 Report

Dear Authors,

After the review process, I have several comments: you should insert numerical data in the abstract; if you used alternative methods of analysis, you should include in the introduction new findings regarding the bioactive potential of functional products and bioavailability of phenolic compounds; you should include comments in the discussion about correlations between microbiota bioactivity and bioavailability of functional compounds as a limitative phase after consumption; also, this type of researches represent a possible future valorization of the study. Best regards!

Author Response

(The authors gave the same response as above.)

Reviewer 3 Report

The manuscript entitled “Horned melon pulp, peel, and seed: New insight into phytochemical and biological properties” from Šovljanski et al., studies the different compounds and biological activity of Cucumis metuliferus. After its reviewing, some modifications must be considered.

The preparation of the extracts should be better explained, as it is a crucial step in the final composition of the extracts.

Following the previous argument, the phytochemical characterization is very superficial. The Folin–Ciocalteau method has been shown not to be a good method to express Total phenolic content because it overestimates these compounds, by also reacting with other non-polyphenolic antioxidant compounds.

The same goes for other assays used in the manuscript, which have become obsolete (DPPH, ABTS).

The number of trials and researched information could contribute to a better understanding of this fruit and its properties, but in the opinion of this reviewer, a better phytochemical characterization should be carried out

Author Response

(The authors gave the same response as above.)

Round 2

Reviewer 1 Report

I think that the re-submitted manuscript entitled “Horned melon pulp, peel, and seed: New insight into phytochemical and biological properties" deserves publication in the Antioxidants after minor revision. Significant positive changes to the manuscript can be seen. Please correct the spelling of the microorganisms. Serotypes is only applicable to specific species microorganisms. The full name is nowhere to be found.

Author Response

Reviewer #1:

I think that the re-submitted manuscript entitled “Horned melon pulp, peel, and seed: New insight into phytochemical and biological properties" deserves publication in the Antioxidants after minor revision. Significant positive changes to the manuscript can be seen. Please correct the spelling of the microorganisms. Serotypes is only applicable to specific species microorganisms. The full name is nowhere to be found.

AUTHORS: The authors would like to thank the Reviewer for a professional review as well as the opportunity to make essential and crucial changes in our work. The reviewer's remark is accepted and the paper is changed according to the comment.

Reviewer 2 Report

No supplementary comments compared with the first review.

Author Response

Reviewer #2:

No supplementary comments compared with the first review.

AUTHORS: Authors want to thank the Reviewer for recognizing the potential of our investigation.

Reviewer 3 Report

After review, the article presents a good quality and, although with low grade experiments, a high number of interesting results.

Author Response

Reviewer #3:

After review, the article presents a good quality and, although with low grade experiments, a high number of interesting results.

AUTHORS: Authors want to thank the Reviewer for recognizing the potential of our investigation.